# Flow analysis on microcasting with degassed polydimethylsiloxane micro-channels for cell patterning with cross-linked albumin

Yigang Shen[1,2][⊘], Nobuyuki Tanaka[1][⊘], Hironori Yamazoe[3], Shunsuke Furutani[3,4], Hidenori Nagai[3,4], Takayuki Kawai[1], Yo Tanaka[1,2]*

**1** RIKEN Center for Biosystems Dynamics Research, Osaka, Japan, **2** Graduate School of Frontier Biosciences, Osaka University, Suita, Osaka, Japan, **3** Biomedical Research Institute, National Institute of Advanced Industrial Science and Technology (AIST), Osaka, Japan, **4** Advanced Photonics and Biosensing Open Innovation Laboratory (PhotoBIO-OIL), AIST, Osaka, Japan

⊘ These authors contributed equally to this work.
* yo.tanaka@riken.jp

**Data Availability Statement:** All relevant data are within the manuscript and its Supporting Information files.

## Abstract

Patterned cell culturing is one of the most useful techniques for understanding the interaction between geometric conditions surrounding cells and their behaviors. The authors previously proposed a simple method for cell patterning with an agarose gel microstructure fabricated by microcasting with a degassed polydimethylsiloxane (PDMS) mold. Although the vacuum pressure produced from the degassed PDMS can drive a highly viscous agarose solution, the influence of solution viscosity on the casting process is unknown. This study investigated the influences of micro-channel dimensions or solution viscosity on the flow of the solution in a micro-channel of a PDMS mold by both experiments and numerical simulation. It was found experimentally that the degassed PDMS mold was able to drive a solution with a viscosity under 575 mPa·s. A simulation model was developed which can well estimate the flow rate in various dimensions of micro-channels. Cross-linked albumin has low viscosity (1 mPa·s) in aqueous solution and can undergo a one-way dehydration process from solution to solid that produces cellular repellency after dehydration. A microstructure of cross-linked albumin was fabricated on a cell culture dish by the microcasting method. After cells were seeded and cultivated on the cell culture dish with the microstructure for 7 days, the cellular pattern of mouse skeletal myoblast cell line C2C12 was observed. The microcasting with cross-linked albumin solution enables preparation of patterned cell culture systems more quickly in comparison with the previous agarose gel casting, which requires a gelation process before the dehydration process.

## Introduction

In the field of cell biology, because the spatial arrangement of cells is an important factor in morphogenesis and tissue functions [1,2], artificially generated spatial patterns of cells have been often used. As an example, when human mesenchymal stem cells are confined in a

**Funding:** This research was supported by RIKEN-AIST"challenge research" project, RIKEN BDR organoid project, JSPS Grant-in-Aid for Scientific Research on Innovative Areas(19H05338), and Sasakawa Scientific Research Grant (20192031). The funders had no role in study design,data collection and analysis,decision to publish,or preparation of the manuscript.

**Competing interests:** The authors have declared that no competing interests exist.

minute area to induce cell differentiation, the shape of the area is such that bone differentiation can be seen near the boundary of the confined area and fat differentiation can be seen in the area distant from the boundary [3–5]. In tissue engineering, myocytes well aligned by a stripe micropattern of polymer brushes have a potential to produce higher contraction than that in the native condition [6,7]. The well-aligned hanging drops of hydrogels have been proposed for forming and imaging three-dimensional tissues [8]. As a cell spatial arrangement technique, the straightforward and inexpensive micro contact printing method has been used, in which a cell adhesive substance such as fibronectin is transferred to a cell culture surface as an ink [9,10]. Although micro contact printing is simple and easy to use, the pattern resolution is limited by the stamp deformation and ink mobility. Microfluidics has been shown to be useful for generating a precise and complex microenvironment to regulate the behaviors of stem and progenitor cells [11–13]. The authors have succeeded in patterning cells using a microstructure cast of agarose, one of the non-cell-adhesive substances [14,15]. In these two studies, a casting solution was driven into a degassed polydimethylsiloxane (PDMS) mold by the accumulated vacuum pressure in the PDMS. The efficiency and simplicity of microstructure preparation can be enhanced by repeatedly using a PDMS mold with a desired pattern without additional liquid driving instruments. Because this vacuum pressure is caused by the diffusion of air into spaces inside PDMS elastomer [16,17], the time in which the vacuum pressure is produced is limited with a time constant of around 10 min [14]. It is known that when a high viscosity solution flows in a narrow channel, it generates a large pressure loss, reducing the flowability. This might cause the introduction of solution to take longer than expected from vacuum production time constant, resulting in insufficient casting of the microstructure. Furthermore, the gelation process of agarose solution, which requires an uncertain time in the micro-channel, is required in agarose microcasting. The authors have also developed a gas-based cell repellent surface patterning method by using the degassed PDMS to suck a special gas into the micro-channel for surface treatment [18,19]. This method could also be used for ultra-small patterning or patterning in a micro-channel [20], which is normally done by a thermal fusion bonding method [21,22]. However, the cell repellency was not so strong compared with gel patterning.

To reduce the complexity of using the agarose casting and add operability of the cell patterning, a low viscosity aqueous solution of cross-linked albumin was introduced as a casting solution and it formed the pattern for cell culture (Fig 1A–1G). In recent years, simple cell patterning has been realized by cross-linking albumin, which is a protein that is abundant in serum, and preparing functional materials that can repel cell adhesion [23,24]. Because the cross-linked albumin solution has a one-way dehydration process from solution to solid and an insoluble property after dehydration, the gelation step required with an agarose solution can be eliminated. As there is a difference in viscosity for agarose (34.6 mPa·s) and cross-linked albumin (1 mPa·s) at 30˚C, this study first investigated the influences of micro-channel dimensions and solution viscosity on the flow of solution in a micro-channel of a PDMS mold by both experiments and numerical simulation. Furthermore, to clarify the durability of cross-linked albumin under cell culture conditions, this study compared the cell patterns of mouse skeletal myoblasts C2C12 with microstructures made of native and cross-linked albumin. Finally, this study compared the cell patterns of mouse skeletal myoblasts C2C12 with microstructures made of native and cross-linked albumin.

## Materials and methods

### Preparation of PDMS micro-molds

The fabrication procedure was based on that of previous papers [14,15]; and PDMS micro-molds were prepared by a photolithographic method. Briefly, negative photoresist (SU-8 2035)

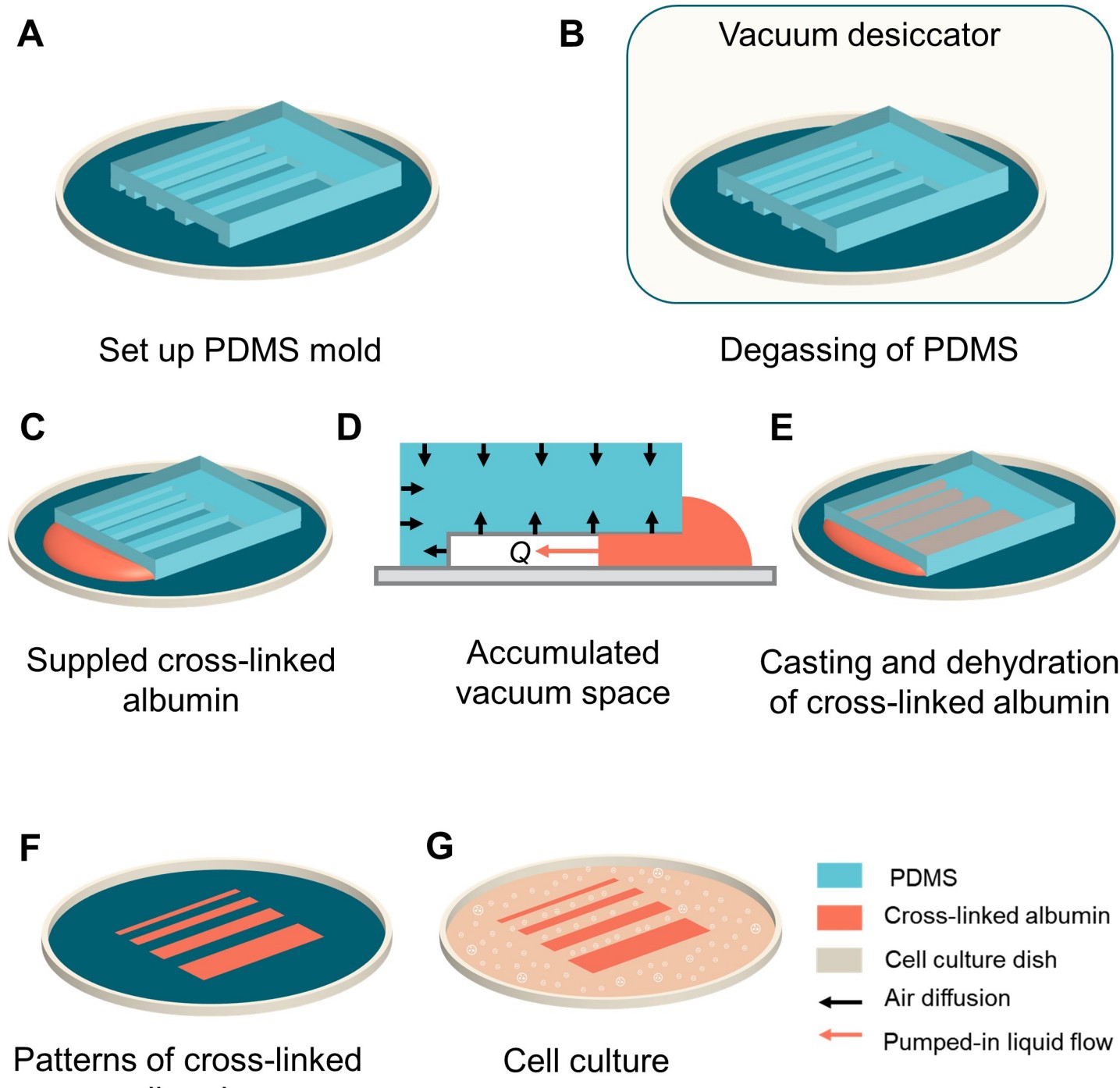

**Fig 1. Schematic illustration of microcasting with cross-linked albumin and application to patterned cell culture.** (A) A PDMS mold with micro-channels was placed on a cell culture dish. (B) The mold on the dish was degassed in vacuum for 1 h. (C) After degassing, cross-linked albumin solution was immediately supplied at the open ends of the micro-channels. (D) The solution was introduced into the micro-channels by the vacuum pressure accumulated in the mold. (E) The micro-channels were filled with cross-linked albumin solution within several minutes. (F) After dehydration of the cross-linked albumin, the mold was carefully removed from the dish, and the micro-channel pattern of cross-linked albumin was obtained on the dish. (G) Cell culture on the dish with the cross-linked albumin pattern.

**Table 1. The conditions of spin coating.**

| Desired thickness (μm) | Photoresist | Initial rotation | | | Final rotation | | | Intensity of UV-light (mW/cm$^2$) | Time of exposing (s) |
|---|---|---|---|---|---|---|---|---|---|
| | | Slope (s) | Speed (rpm) | Duration (s) | Slope (s) | Speed (s) | Duration (s) | | |
| 30 | SU8 3050 | 5 | 1500 | 10 | 10 | 3000 | 20 | $9.89 \times 10^3$ | 12 |
| 100 | SU8 100 | 5 | 500 | 10 | 10 | 3000 | 20 | $9.89 \times 10^3$ | 15 |
| 200 | SU8 2100 | 5 | 500 | 10 | 5 | 1500 | 25 | $9.11 \times 10^3$ | 40 |
| 400[a] | SU8 2100 | 5 | 500 | 10 | 10 | 3000 | 20 | $9.11 \times 10^3$ | 53 |
| 800 [a] | SU8 2100 | 5 | 500 | 10 | 3 | 1000 | 27 | $9.11 \times 10^3$ | 100 |

[a] The process used double layers of photoresist formed on the Si-wafer substrate.

(Nippon Kayaku, Tokyo) was coated on a 2-inch-diameter Si-wafer (p-type, mirror-finished <100> surface) (SEMITEC, Chiba) with a spin-coater (1H-D7) (Mikasa, Tokyo) spinning at specific conditions. The rotation program of the spin-coater was modified to make different heights (30, 100, 200, 400 and 800 μm) of the PDMS micro-channel model (Table 1). The photoresist covering the Si-wafer was baked at 65˚C for 3 min and 95˚C for 5 min. The photoresist was exposed to UV-light at the specified intensity and time (Table 1) using a mask aligner (MA-10) (Mikasa) through a photomask having a desired pattern. After UV-exposure, the Si-wafer with the photoresist pattern was baked at 65˚C for 1 min and 95˚C for 5 min, and subsequently treated with a developer, 2-methocy-1-methylethyl acetate (130–10505) (Wako, Osaka). The Si-wafer and its remaining photoresist were baked at 150˚C for 5 min and used as a mold for the micro-molds. PDMS and a curing agent for PDMS (Sylpot 184 W/C) (Dow Corning Toray, Tokyo) were mixed at a ratio of 10 to 1. This PDMS mixture was poured into the Si-wafer mold and cured at 80˚C for 3 h. The cured PDMS was peeled from the Si-wafer and trimmed, and a micro-mold for casting microstructures was obtained.

## Preparation of cross-linked albumin

Cross-linking of albumin was done as described previously [24]. Bovine serum albumin (A6003) (Sigma-Aldrich, MO) was dissolved in phosphate-buffered saline (PBS, pH 7.4) to yield a 3 w/v% solution, which was then reacted with 215 mM ethylene glycol diglycidyl ether (EGDE) (056–03841) (Wako) with vigorous stirring for 24 h at 25˚C. The reaction mixture was dialyzed for 3 days at room temperature against Milli-Q water using cellulose tubing (molecular weight cutoff = 12 kDa) (Nihon Medical Science, Gunma) to remove the unreacted EGDE. The reaction mixture was then adjusted to a final concentration of 2% by adding Milli-Q water, sterilized by filtering through a 0.22-μm filter, and stored at 4˚C.

## Preparation of cell culture substrate with albumin microstructures

Native or cross-linked albumin microstructures were fabricated on a tissue culture polystyrene (TCPS) dish (Falcon 353003) (Corning, Inc., NY) by the degassed PDMS assisted microcasting method with albumin solution instead of the agarose one, which has been used in the authors' previous studies [14,15]. First, the surface of the fabricated PDMS mold was cleaned using mending tape (MP-18) (Sumitomo 3M, Tokyo). After being rinsed with ethanol and deionized water, the PDMS mold was placed on the bottom surface of the TCPS dish; care was taken to not have any bubbles between the mold and surface. The dish with the mold was degassed in a vacuum desiccator (VL-C) (ASONE, Osaka), which was connected to a vacuum line at a gauge pressure of -98 kPa for 1 h degassing. After being degassed, the dish with the mold was immediately removed from the desiccator and the solution of native or cross-linked albumin was applied onto the open

ends of the micro-channels of the mold within several minutes. The behavior of solution flowing in the channels was recorded by a digital video camera (HDR-CX560) (Sony, Tokyo). After dehydration of the solution, the mold was removed from the dish. Finally, the surface profiles of the dehydrated cross-linked albumin microstructure were observed with a three-dimensional laser scanning confocal microscope (VK-8710) (Keyence, Osaka).

### Cell culture

Mouse skeletal myoblast (C2C12) (RCB0981) (Riken BRC cell bank) cell line was used for the cell culture. First, Dulbecco's modified Eagle's medium (D6429) (Sigma-Aldrich) supplied with 10 v/v% fetal bovine serum (Nichirei Biosciences, Tokyo) and 1 v/v% penicillin–strepto-mycin (168–23191) (Wako) was prepared to maintain the cells. The cells were passaged by a trypsin-digest method before full-confluency. To demonstrate patterned cell culturing, the cells were seeded onto two dishes with the micro-cast structures of native albumin and cross-linked albumin at an initial density of $2.6×10^4$ cells/cm$^2$. Finally, the cells were cultured at 37°C in a humidified condition with 5% $CO_2$.

### Viscosity, surface tension, and contact angle measurement

The viscosity of cross-linked albumin solution was measured with a viscosity meter (SV-10) (A&D, Tokyo). In order to study the influence of viscosity on the flow rate caused by the degassed PDMS, different average molecular weights of polyethylene glycol (PEG) (6000, 4000 and 600) (169–09125, 162–09115 and 168–09075) (Wako) were used to prepare viscous solutions, which were also subjected to the viscosity measurement at room temperature. The behavior of prepared solution flowing in a micro-channel of the PDMS mold was also recorded by the video camera and the flow rate of solution was analyzed by video analysis open source software (Tracker version 5.1.3 on Windows) [25]. The surface tension and contact angles of solution against PDMS or TCPS surfaces were measured by a contact angle meter (DMe-401) (Kyowa Interface Science, Saitama, Japan) and drop shape analysis software (FAMAS) (Kyowa Interface Science).

### Microscopic observation

An inverted phase-contrast microscope (IX73) (Olympus, Tokyo) was used for microscopic observation. The cultivated cells were observed every 24 h. The cross-linked albumin coated areas and bare-surface exposed areas were manually determined from the microphotographs. After the beginning of cell culture, the areas of presence and absence of cells were automatically determined from the microphotographs by using ImageJ software (S1 Text and S1 Fig).

## Results and discussion

### Cross-linked albumin solution casting

A PDMS mold with the designed structure was fabricated by the photolithography method (Fig 2A) and cross-linked albumin supplied at the open ends of the micro-channels was draw into the micro-channels. After drying at room temperature for at least 4 h, the PDMS mold was peeled from the dish surface and the dehydrated cross-linked albumin pattern of micro-channels was obtained on the dish surface (Fig 2B). The microstructure on a cell culture surface was confirmed under the confocal laser microscope (Fig 2C). When the shape of the microstructure was measured with this microscope, there was a wall structure near the boundary between the microstructure and the cell culture dish (Fig 2D). This was assumed to be derived from the meniscus between the side wall of the PDMS mold and bottom surface,

which was formed during the dehydration process. Furthermore, the existence of this structure on the dish surface suggested the cross-linked albumin structure had a higher adherent force onto the dish than onto the PDMS mold. The cross-linked albumin structure was significantly

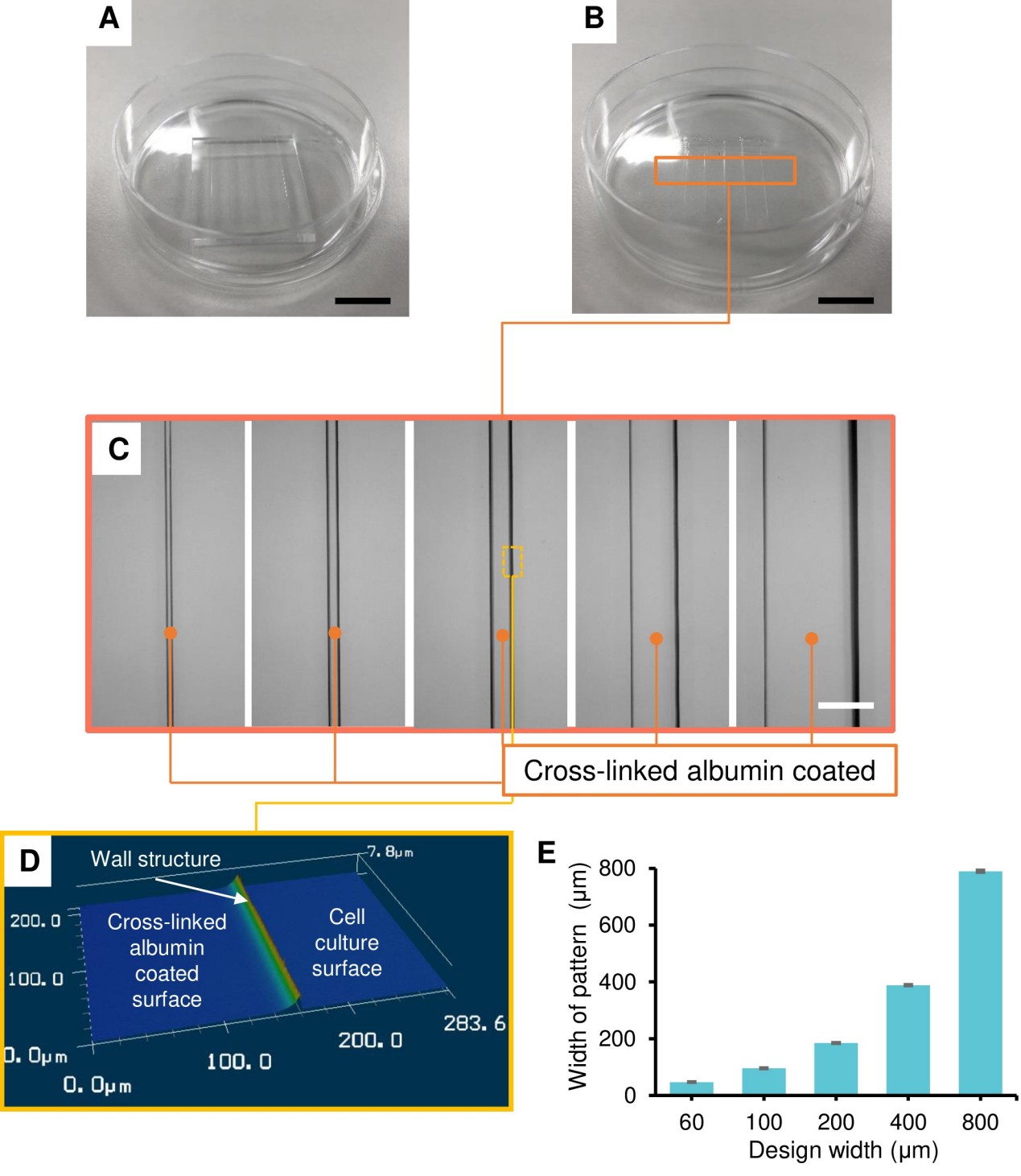

**Fig 2. Microstructure of cross-linked albumin.** (A) PDMS micro-mold (height: 200 μm) and (B) the microstructure on a cell culture surface. (C) Microphotographs and (D) three-dimensional shape of the microstructure of the cross-linked albumin pattern; the black bar indicates 1 cm and the white bar indicates 400 μm. (E) The graph shows the width of the cross-linked albumin pattern (mean±SD, n = 5).

shrunk in the height direction in comparison to the structure height of the previously obtained agarose pattern [14]. On the other hand, the widths of cross-linked albumin patterns were slightly decreased compared with the PDMS mold (Fig 2E).

## Mechanism and simulation of degas-driven flow

Regarding spatial controllability for cell patterning, various geometry structures need to be formed in the culture dish. Also, these patterns should be prepared in areas with widths from millimeter to centimeter order in the case of large-area casting [15]. Therefore, the value of the flow rate would be useful for estimating the time needed for the casting process. To estimate the flow rate of liquid in the micro-channel, this study employed a one-dimensional model (Fig 3A) with the three influencing physical conditions: (1) the air solubility of PDMS, (2) the friction of liquid and (3) the capillary effect. As air diffuses into the PDMS, a pressure difference (vacuum pressure) is generated in the micro-channel relative to the atmospheric pressure. This vacuum pressure is the main driving force, which relates to features of PDMS, such as solubility, diffusivity and porosity. According to Henry's law, the gas solubility in PDMS is described as:

$$S = \frac{C}{p} \tag{1}$$

where $S$ [cm$^3$ (standard conditions for temperature and pressure, STP)/(cm$^3$ atm)] is the solubility coefficient and $C$ [cm$^3$ (STP)/ cm$^3$] is the equilibrium gas concentration in PDMS at pressure p [atm]. It is known that the solubilities of $O_2$ and $N_2$ in PDMS are independent of the penetrant pressure but have a positive linear relationship with the temperature [26]. The ability for gas transfer depends on diffusivity in PDMS. Based on Fick's law of diffusion, the air flux F at the surface of degassed PDMS can be described as:

$$F_{air} = D\frac{\partial C}{\partial t}\bigg|_{x=l} \approx \frac{D_0(C_1 - C_0)}{l}\exp\left(-t\frac{\pi^2 D_0}{4l^2}\right) \tag{2}$$

where $D$ is the diffusion coefficient of gas in PDMS, $D_0$ is the average diffusion coefficient of gas in PDMS, $C_1$ is air concentration in PDMS before degassing, $C_0$ is the concentration of air inside PDMS after degassing, $C_0 = C(x, 0)$ is the equilibrium concentration in the degassing step, and $l$ is the thickness of the PDMS mold model. Therefore, high gas solubility and the actual gas diffusion material can lead to a high pressure difference and larger air flux in the degassed PDMS. When it comes to the porosity, the gas-dissolving volume and accumulation recovery flux are assumed to be increased with the pore size and distance between neighboring pores [27]. On the other hand, the high flux may cause a shortened operable time for driving the solution. Therefore, the optimal porosity should be considered from the viewpoint of practical use.

This study hypothesized that the mechanism of driving liquid is that of air diffusion through the micro-channel surface into the PDMS mold and pressure-driven Hagen–Poiseuille-like fluid flow within micro-channels, and that the fluid will replace the decreasing air flux in time. Hence, the volumetric flow rate $Q$ can be described by the following equation

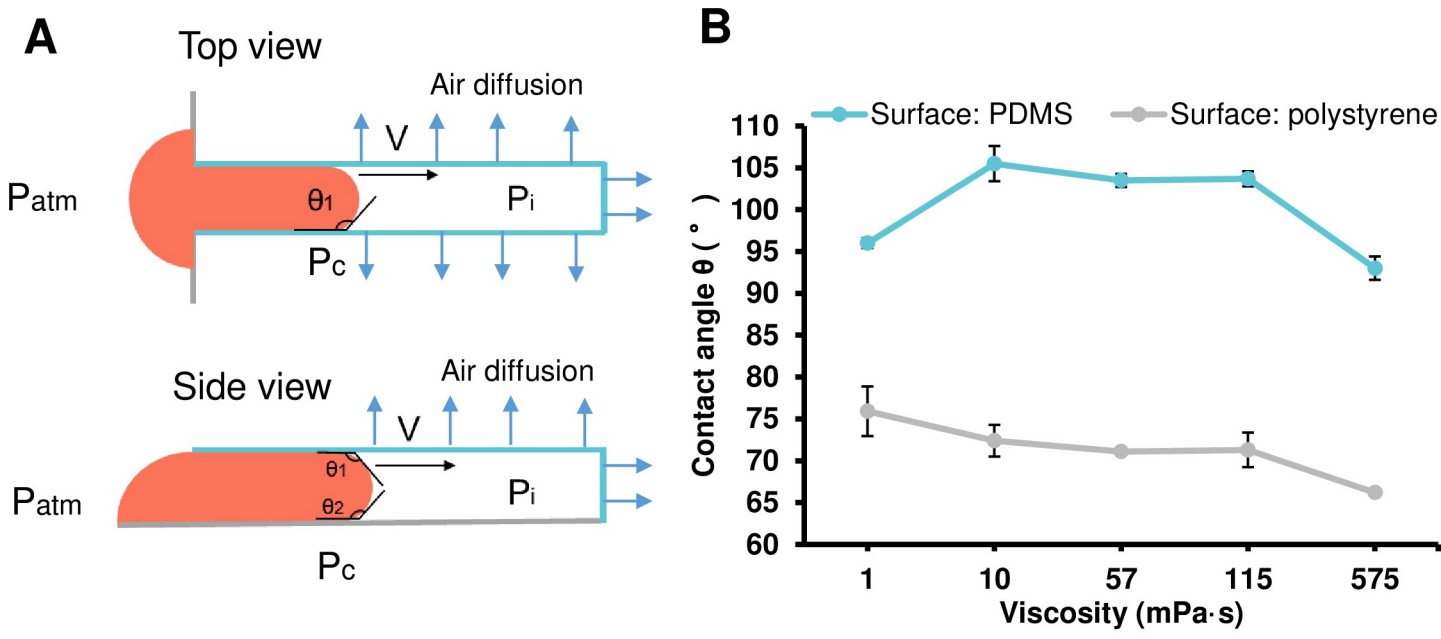

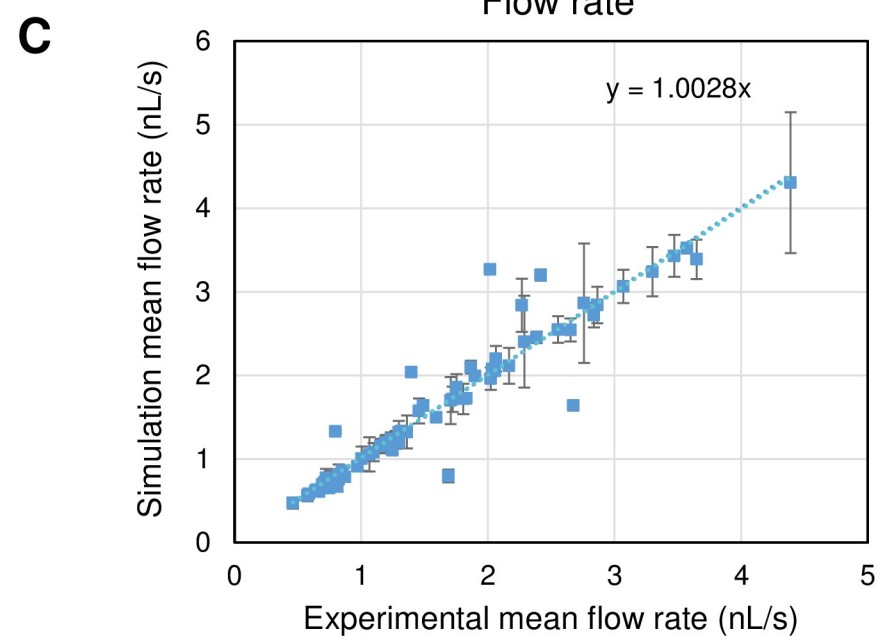

**Fig 3. Simulation model for casting in the degassed PDMS dead-end channel.** (A) Pressures in the degassed channel. $P_{atm}$ is the atmosphere pressure; $P_i$ is the air pressure inside the channel; $P_c$ is the pressure caused by capillary force; $\theta_1$ and $\theta_2$ are the dynamic contact angles of the fluid with, respectively, the wall and the bottom material of the channel. (B) Graph showing contact angle of agarose solution of different viscosity. Data points and error bars indicate mean±SD (N = 5). (C) Image showing the correlation between simulation and experimental data.

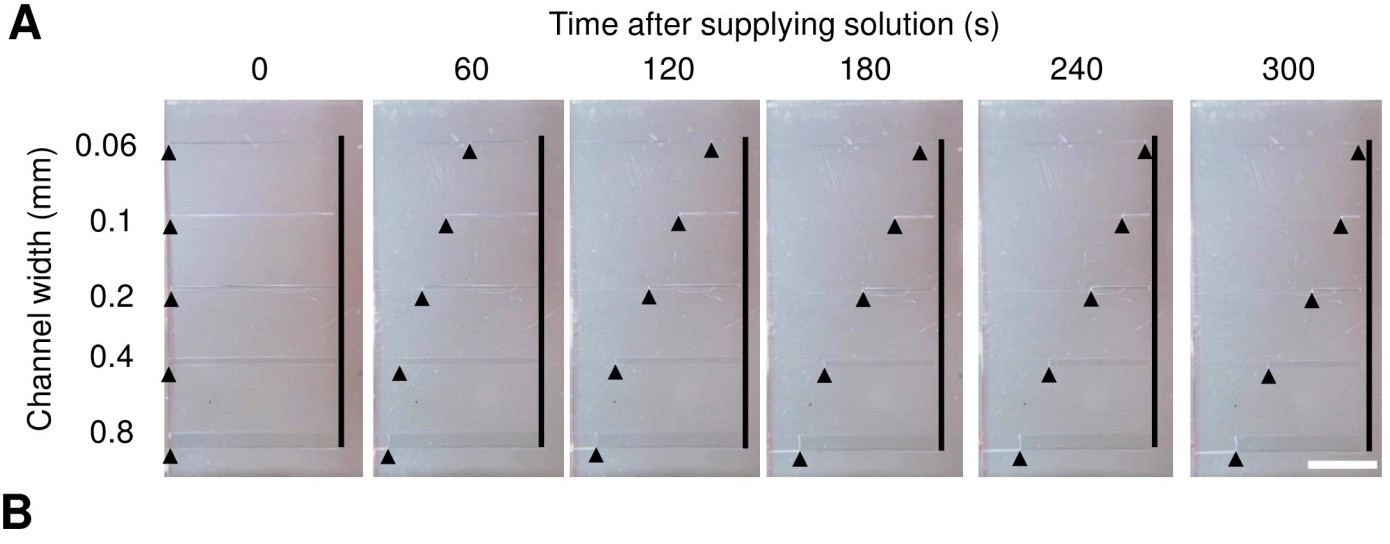

**Fig 4. Photographs showing the appearance of cross-linked albumin solution and graphical time-courses of flow rates.** (A) Appearance of cross-linked albumin solution (1 mPa·s) at 0, 30, 60, 120, 180, 240, and 300 s after supplying the solution at the end of the 1 mm-length micro-channels (height: 0.2 mm) of degassed PDMS molds with accumulated vacuum pressure. Arrow heads and lines indicate the head points of cross-linked albumin solution penetrating into the micro-channels and

the dead-end of micro-channels, respectively. The white bar indicates 5 mm. (B) The graph shows time-courses of flow rates with five different channel widths after supplying the solution at the end of the 1 mm-length micro-channels (height: 0.2 mm) of degassed PDMS molds with accumulated vacuum pressure; data points and error bars indicate mean±SD (N = 3). The solid colored lines corresponding to the data points are simulation results with the same dimensional conditions of the micro-channel.

[28,29]

$$Q \propto F_{air} = k \frac{A(t)D_0(C_1 - C_0)}{l} \exp\left(-t\frac{\pi^2 D_0}{4l^2}\right) \tag{3}$$

$$Q = \frac{\Delta P}{R_f} \tag{4}$$

where $F_{air}$ is the air flux, $k$ is an empirical factor related to the wetting property and capillary effect of solution. $A(t)$ is the surface that air diffuses through. $R_f$ is the flow resistance caused by the friction of liquid and $\Delta P$ is the pressure difference between pressure in a micro-channel ($P_i$) and atmospheric pressure ($P_{atm}$).

$$\Delta P = P_{atm} - P_i + P_c \tag{5}$$

$P_c$ is the capillary pressure caused on the interface of the solution in a rectangular cross-sectional micro-channel and it is given by the following equation [30]

$$P_c = \cos\theta_c \gamma \left(\frac{2}{w} + \frac{1}{h}\right) + \cos\theta_b \gamma \frac{1}{h} \tag{6}$$

where $\theta_c$ is the contact angle (CA) of the solution with the micro-channel wall; $\theta_b$ is the CA of the solution with the micro-channel bottom (Fig 3B); $\gamma$ is surface tension of the solution; $h$ and $w$ are height and width of the micro-channel, respectively. From the model, ordinary differential equations were constructed and were solved by home-made software (S2 Text and S2 File). The results in Fig 3C indicated the regression equation of the line was y = 1.0028x, which was accepted as a very high correlation.

## Flow characterization in the casting process

The appearance of cross-linked albumin solution after supplying the solution at the open ends of the 1 mm-length micro-channels (height: 0.2 mm) of PDMS molds can be seen in Fig 4A. When combined with the results from Fig 4B, it was concluded that even the wider channel had a faster flow rate during 5 min, and the time needed to fill the entire channel with liquid became longer as the channel dimensions were increased. Fig 4B indicated that the flow rates were unable to increase to compensate for the increase in cross-sectional area, resulting in an increase in casting time for the wider channel. And there was a significant difference for 0.06 mm and 0.8 mm width channels between the experiment and simulation; this was because the experiment and simulation model assumed a continuous flow in the casting process, but for these channel widths, the flow had stopped in the middle of the experiment or it moved at too slow a speed to be observed.

From the corresponding analysis between structure parameters (surface area, width and height) and mean flow rate, solution viscosity and mean flow rate in both experiment (Fig 5A–5D) and simulation (Fig 5E and 5F), it was demonstrated that the surface area had a high correspondence with the flow rate. When one parameter (width or height) was fixed, the height parameter had a bigger effect on the flow rate compared to the width parameter. Since the main driving force in the micro-channel filling was produced by air diffusion on the surface of

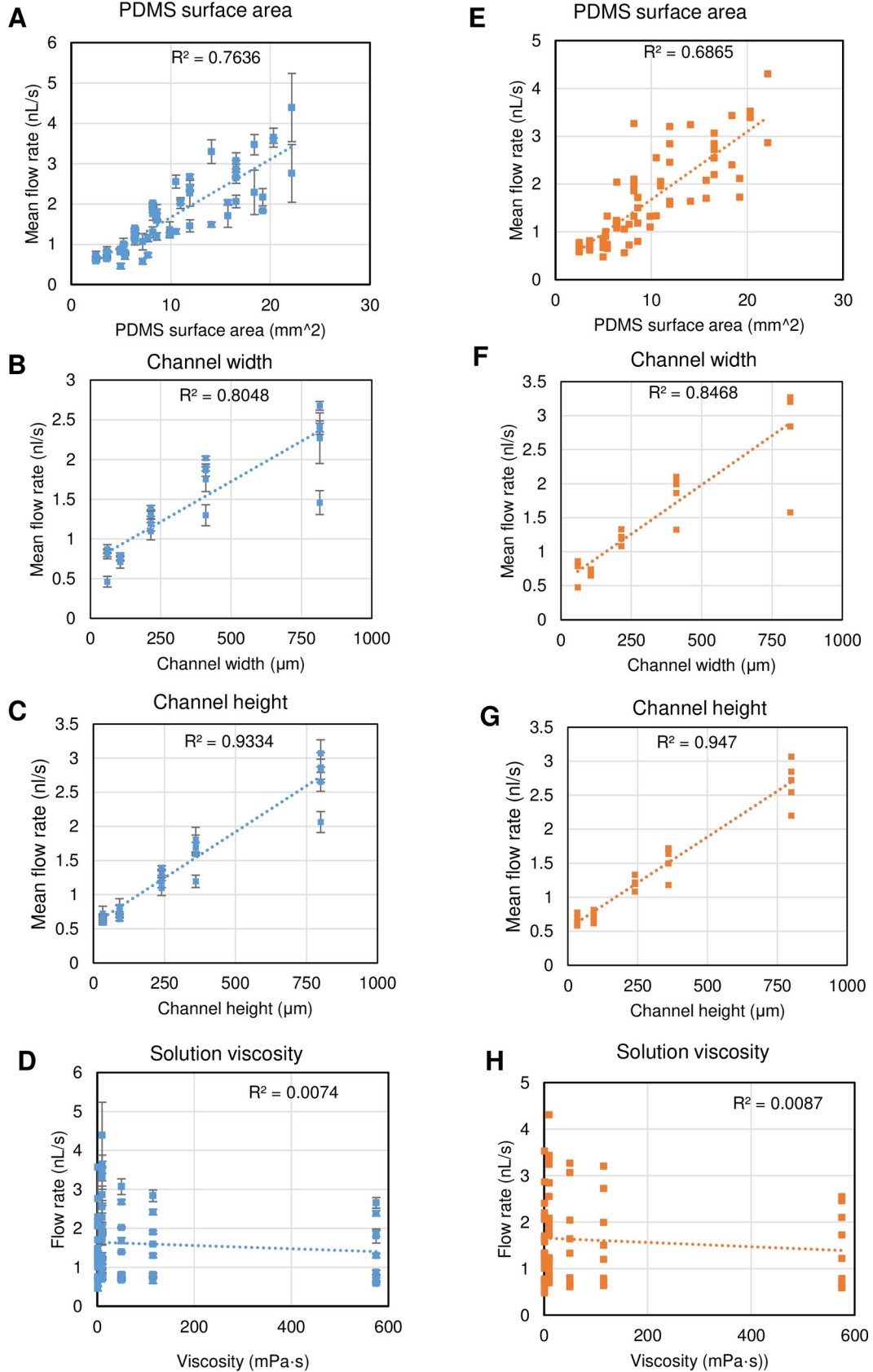

**Fig 5. Relationships between structure parameters (surface area, width and height) and mean flow rate, solution viscosity and mean flow rate.** (A)-(D) Images showing the experimental data; data points and error bars indicate mean±SD (N = 3). (E)-(H) Images showing simulation data.

the degassed PDMS, by increasing the height and width to the same degree, the height increase would lead to greater surface area changes. However, the low coefficient of determination $R^2$ (0.0074 for experimental data and 0.0087 for simulation data) indicated that there was no correspondence between viscosity and flow rate. In other words, this liquid with a wide range of viscosity from 1 to 575 mPa·s could be driven into the micro-channel with little change of flow rate. As the pressure difference $\Delta P$ dominated the flow rate, it was assumed that the small variation of pressure at different viscosity caused the viscosity-independent tendency. This is described in more detail in the section "Evaluation of viscosity effect for the flow rate".

## Evaluation of viscosity effect for the flow rate

To understand the viscosity-independent tendency on the flow rate of solution flowing in a micro-channel, it is critical to clarify the pressure change in the casting process. Because the driving pressure is the difference between the ambient pressure outside the mold and vacuum pressure in cavities of the mold, normally the driving vacuum pressure is a maximum of 1 atm. It is not possible to use a higher pressure than atmospheric pressure to generate a high flow rate in this method. However, it was hard to measure the actual pressure in the micro-channel, and so the simulation model was used to analyze this issue. The pressure loss caused by the viscosity increased dramatically from 1 to 575 mPa·s (Fig 6A). Actually, such a pressure loss will cause an obvious decrease in normal flow conditions. From the simulation results, the vacuum pressure in the micro-channel could be estimated. The time-series of vacuum pressure was found to vary with the values of solution viscosity. And the maximum vacuum pressure was from 30 to 40 KPa at 40 to 60 s after solution application, which was over 10 times higher than the pressure loss at least (Fig 6B). Therefore, the slight change of vacuum pressure was one of the reasons for occurrence of the viscosity-independent tendency. This tendency, which has not been reported in other related research studies [31–34], is mainly caused by the dead-end degassed PDMS channel. In this structure, the exposed inner surface area of the PDMS micro-channel, producing the vacuum pressure, is decreased by flow of the casting solution, and by contrast, the contact area is increased between the solution and PDMS wall, causing the viscous resistance. Reducing the exposed inner area causes a bottleneck for obtaining the driving pressure, resulting in the diminishing of both flow rate and pressure loss. Therefore, the decrease of the inner surface area was speculated to influence the flow rate of casting solution relatively more than the solution viscosity did.

## Micro-patterning of cells

Mouse skeletal myoblasts C2C12 adhered on the entire surface of the cell culturing dish with the native albumin microstructure for the 1-d cultivation (Fig 7A and 7B). The shape of the cells was elongated relative to the surrounding cells; however, there was no orientation tendency. On the other hand, cell adhesion was clearly blocked by the cross-linked albumin microstructure for at least 7 days (Fig 7C–7F). This suggested that the cross-linked albumin had a potential for micro-patterning cells for long-time cultivation as a negative patterning method based on the sufficient repellency of the dehydrated cross-linked albumin against adherent cells such as C2C12. Although the center part of the cross-linked albumin structure was quite thin and hard to observe by confocal microscopy (Fig 2D), this thin structure still maintained good repellency to the cell adhesion during the 7-d cultivation. Because the native

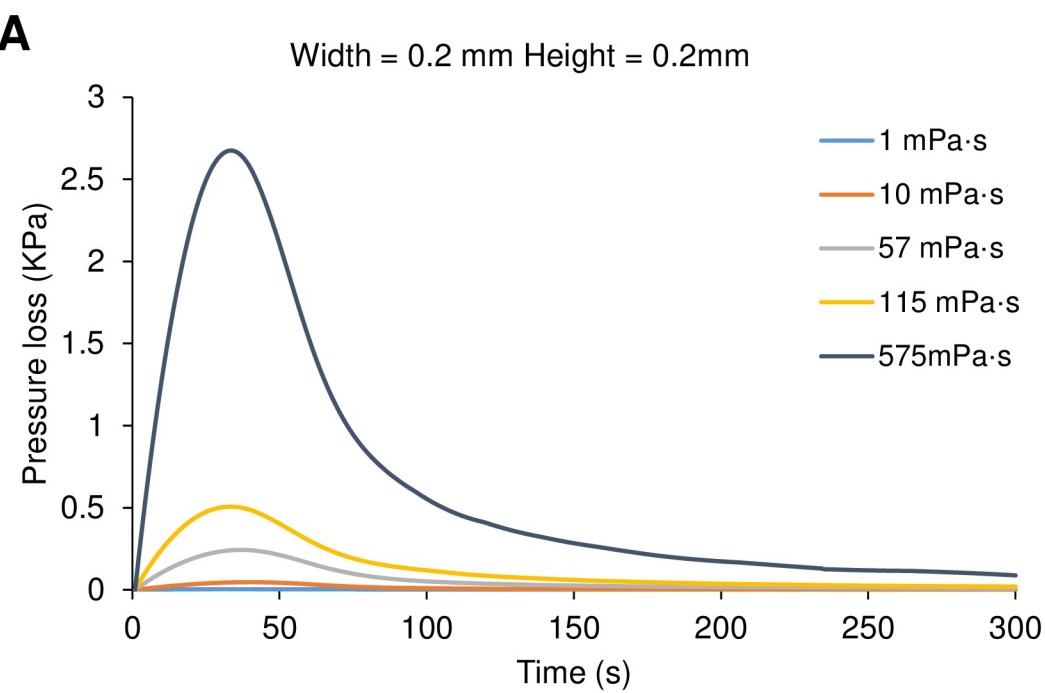

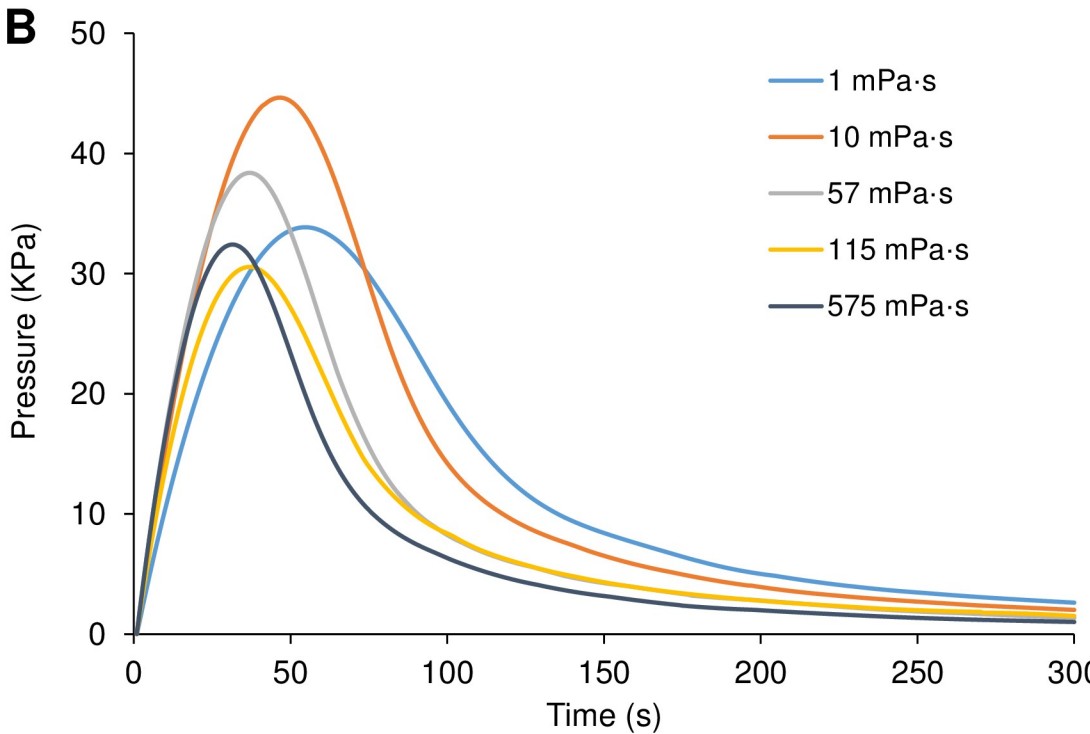

**Fig 6. Pressure analysis of polyethylene glycol solution.** (A) and (B) show estimated time-courses of pressure loss caused by viscosity and application time of the internal vacuum pressure, respectively (simulation results).

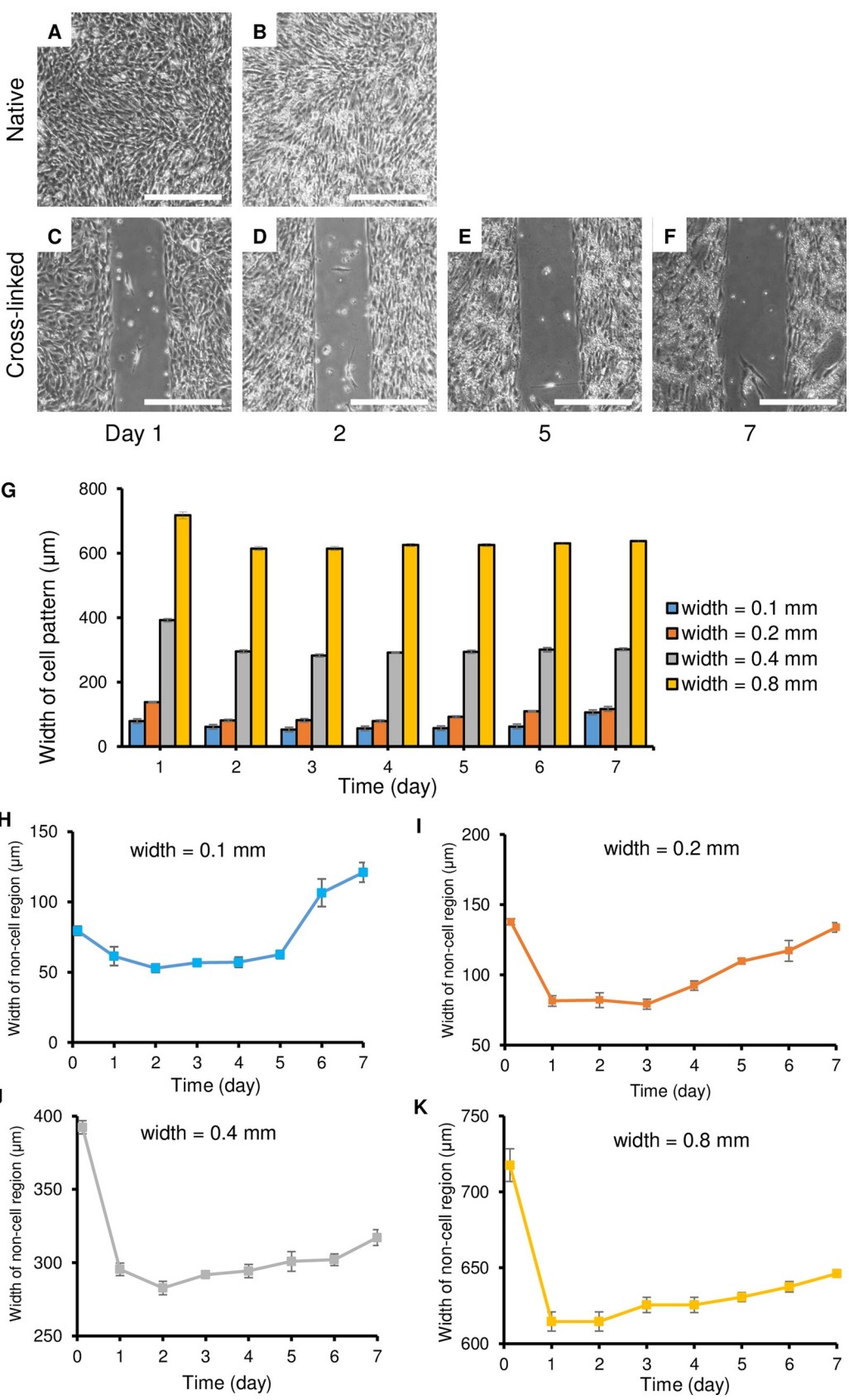

**Fig 7. Adhesion-area-regulated cell culture with two different albumin microstructures cast with 0.4-mm-width and 0.2-mm-height micro-channels.** (A) and (B) Microphotographs showing mouse skeletal myoblasts C2C12 adhered on a surface with the native albumin structure at 1 and 2 d, respectively. (C), (D), (E), and (F) Microphotographs showing myoblasts with the cross-linked albumin at 1, 2, 5, and 7 d, respectively. White scale bars indicate 400 μm. (G) The variation of width in patterns without cells during one week (mean±SD, n = 5). (H-K) Relationships between the designed width and albumin structure width and no-cell region width during one week (mean±SD, n = 5).

albumin has a high water-solubility, it was speculated that the replacement of molecules on the dish surface was caused between native albumin and other proteins derived from the serum. As previously reported [35,36], serum contains major cell adhesion proteins, fibronectin and vitronectin, and they are anchors for cell adhesion. On the other hand, both the low solubility of the cross-linked albumin and its adherent strength to the dish suppressed the detachment of molecules and attachment of serum proteins, especially with cell adhesion function.

After the 2-d cultivation, the width of the cell-repellent region was found to be decreasing (Fig 7C–7F). This suggested that the repellency of the cross-linked albumin had changed with time. Therefore, the influence of the time duration of cell culturing and geometrical dimensions of the pattern for the repellency ability needed to be observed (Fig 7G–7K and Supporting information S1 Text). The difference in width between the cross-linked albumin pattern and the real cell pattern decreased and then slightly increased (Fig 7H–7K). The pattern (width = 0.06 mm) lost its repellency to cells after the 1-d cultivation (Fig 7B), and the cells grew all over the area which formed the cross-linked albumin. For the larger pattern (width > 0.1mm) repellency ability was kept for a week. Indeed, there were some cells that grew in an area of the cross-linked pattern caused by casting defects. From the area ratio of casting defects (S1 Text and S2 Fig), the maximum ratio was measured below 15%, and it could be optimized by modifying the geometrical dimensions of the pattern These results suggested that the cross-linked albumin had a high potential to regulate cellular patterns even for a 7-d cultivation (in other words, the pattern could remain for at least one week). The geometrical dimensions of the pattern influenced the repellency ability of the cross-linked albumin; that is, a narrow structure was unstable for cell culturing and had less repellency.

## Conclusion

Through experiments and numerical simulations, this study analyzed the solution flows driven by vacuum pressure in rectangular cross-sectional shaped degassed PDMS dead-end micro-channels. This microcasting method enabled filling of the micro-channels with solutions of various viscosities from 1 to 515 mPa·s. Within this viscosity range, the pressure loss due to the viscosity, which was the dominant factor, was smaller than the vacuum pressure as a driving force, and the flow rate was almost unaffected by the viscosity. Cross-linked albumin with a viscosity around 1 mPa·s was used as a casting solution, microstructures were fabricated with fewer fabrication steps as compared with the previous agarose approach. This cross-linked albumin microstructure could be used for controlling cell adhesion just as for agarose patterns. Therefore, the degassed PDMS microcasting of cross-linked albumin solution would be useful as a simplified micro-fabrication approach to provide a micro-environment for adhesion-area controlled cell culture systems.

## Supporting information

**S1 Text. A method for determining the area of cell attachment.**
(DOCX)

**S2 Text. Numerical computation of liquid flow in micro-casting with a degassed PDMS mold.**
(DOCX)

**S1 File. Original images of cell culture for imaging processing.**
(ZIP)

**S2 File. A Python program for solving ordinary differential equations.**
(ZIP)

**S1 Dataset. All data include the measured flow rate, contact angle, surface tension, dimensions of PDMS mold, simulation analysis, cell culture and data analysis of cell culture.**
(XLSX)

**S1 Fig. An example of the image process that converted the original image after the 1-d cell culture.**
(DOCX)

**S2 Fig. Area ratio of non-specific cell adhesion region and cell-blocking region in a one-week cell culture.**
(DOCX)

**S1 Table. Parameters of equations.**
(DOCX)

**S2 Table. Value of parameters in the model.**
(DOCX)

## Author Contributions

**Conceptualization:** Nobuyuki Tanaka, Hironori Yamazoe.

**Data curation:** Yigang Shen, Nobuyuki Tanaka.

**Formal analysis:** Yigang Shen, Nobuyuki Tanaka.

**Funding acquisition:** Nobuyuki Tanaka, Hironori Yamazoe, Shunsuke Furutani, Hidenori Nagai, Takayuki Kawai, Yo Tanaka.

**Investigation:** Yigang Shen, Nobuyuki Tanaka.

**Methodology:** Nobuyuki Tanaka, Hironori Yamazoe.

**Project administration:** Nobuyuki Tanaka, Hidenori Nagai, Yo Tanaka.

**Resources:** Yigang Shen, Nobuyuki Tanaka, Hidenori Nagai.

**Software:** Yigang Shen.

**Supervision:** Hidenori Nagai, Yo Tanaka.

**Validation:** Nobuyuki Tanaka, Hironori Yamazoe, Shunsuke Furutani, Hidenori Nagai, Takayuki Kawai, Yo Tanaka.

**Visualization:** Yigang Shen, Nobuyuki Tanaka.

**Writing – original draft:** Yigang Shen, Nobuyuki Tanaka, Hironori Yamazoe.

**Writing – review & editing:** Nobuyuki Tanaka.

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
