## [Decision Letter · Decision Letter 0]

7 Jan 2020

PONE-D-19-34244

Flow analysis on microcasting with degassed polydimethylsiloxane micro-channels for cell patterning with cross-linked albumin

PLOS ONE

Dear Dr. Tanaka,

Thank you for submitting your manuscript to PLOS ONE. After careful consideration, we feel that it has merit but does not fully meet PLOS ONE’s publication criteria as it currently stands. Therefore, we invite you to submit a revised version of the manuscript that addresses the points raised during the review process.

We would appreciate receiving your revised manuscript by Feb 21 2020 11:59PM. To enhance the reproducibility of your results, we recommend that if applicable you deposit your laboratory protocols in protocols.io, where a protocol can be assigned its own identifier (DOI) such that it can be cited independently in the future. For instructions see: http://journals.plos.org/plosone/s/submission-guidelines#loc-laboratory-protocols

We look forward to receiving your revised manuscript.

Kind regards,

Talib Al-Ameri, Ph.D

Academic Editor

PLOS ONE

Journal Requirements:

2. We note that Figure 1 in your submission contain copyrighted images. All PLOS content is published under the Creative Commons Attribution License (CC BY 4.0), which means that the manuscript, images, and Supporting Information files will be freely available online, and any third party is permitted to access, download, copy, distribute, and use these materials in any way, even commercially, with proper attribution. For more information, see our copyright guidelines: http://journals.plos.org/plosone/s/licenses-and-copyright.

1.         You may seek permission from the original copyright holder of Figure(s) [#] to publish the content specifically under the CC BY 4.0 license.

3. In your Methods section, please provide additional details regarding the cell lines used in your study and ensure you have described the source. For more information regarding PLOS' policy on materials sharing and reporting, see https://journals.plos.org/plosone/s/materials-and-software-sharing#loc-sharing-materials, and for more information on PLOS ONE's guidelines for research using cell lines, see https://journals.plos.org/plosone/s/submission-guidelines#loc-cell-lines.

"This research was supported by RIKEN-AIST "challenge research" project, RIKEN BDR organoid project, JSPS Grant-in-Aid for Scientific Research on Innovative Areas (19H05338), Sasakawa Scientific Research Grant (20192031) and Japanese Government Scholarship (Ministry of Education, Culture, Sports, Science, and Technology)."

"This research was supported by RIKEN-AIST"challenge research" project, RIKEN BDR

organoid project, JSPS Grant-in-Aid for Scientific Research on Innovative

Areas(19H05338), and Sasakawa Scientific Research Grant (20192031).The funders

had no role in study design,data collection and analysis,decision to publish,or

preparation of the manuscript."

Reviewers' comments:

Reviewer's Responses to Questions

**Comments to the Author**

1. Is the manuscript technically sound, and do the data support the conclusions?

Reviewer #1: Yes

Reviewer #2: Yes

Reviewer #3: Partly

2. Has the statistical analysis been performed appropriately and rigorously? 

Reviewer #1: Yes

Reviewer #2: Yes

Reviewer #3: Yes

3. Have the authors made all data underlying the findings in their manuscript fully available?

Reviewer #1: Yes

Reviewer #2: Yes

Reviewer #3: Yes

4. Is the manuscript presented in an intelligible fashion and written in standard English?

Reviewer #1: Yes

Reviewer #2: Yes

Reviewer #3: No

5. Review Comments to the Author

Reviewer #1: With the pressure-driven pumping mechanism via micro channels provided by a degassed PDMS mold, the authors demonstrated, with rigorous analysis on experimental data and simulation results, a negative patterning methodology using cross-linked albumin that prevents cell adhesion. The authors also investigated the flow rate as functions of channel geometries and solution viscosity, as well as carried out simulations to better understand solution flow mechanisms. All experimental data and simulation results are made available via support information. The well-written manuscript provides a simple and useful approach for patterned cell cultivation by use of the pressure difference provided by degassed PDMS, which is expected to serve as a tool for researchers in biological science.

Before acceptance for publication, however, the following comments should be reflected in a revised manuscript.

(a) The pumping mechanism is due to pressure differences generated by degassed PDMS, which relies on the material's porosity and air solubility. The authors perhaps should present a brief discussion on this aspect in their manuscript.

(b) In section "Evaluation of viscosity effect for the flow rate", the authors assumed pressures of 30 to 40 kPa for their simulation. What are the grounds for assuming such high pressures? In fact, in Ref 14 the authors showed that the "remaining pressure" in degassed PDMS that was exposed to atmosphere pressure was less than 5 kPa within an hour. The authors perhaps want to reconsider their assumption on the pressure, or, at least, need to justify their assumption on the pressures they adopted in their simulation.

Reviewer #2: This paper was focusing on Flow analysis on microcasting with degassed polydimethylsiloxane micro-channels. It is an quite interesting area about biomaterials. The author also used the quite hot topic of patterning. It would arouse of interests of audience. However some points should be adreessed:

1) why the materials of agarose was used?

2)The influence of solution viscosity on the casting process should be further detailed.

3) The introduction should add more information of the purpose.

4) In Fig.5 d and h, the R value is very low, the author should give more explanations.

Reviewer #3: The manuscript applied degassed PDMS’s vacuum pressure property for fabrication of patterned cross-linked albumin, which demonstrate the control on the cell culturing alignment. The results are fairly good for being considered by this journal. However, following points should be addressed: ‘

1. English proofreading is mandatorily needed:

Line 3, Page 3 – “… has …” should be “… have …”

Line 8, Page 3 – what is exactly “surrounding environments”?

Line 10 – Line 12, Page 3 – this micro contact printing technique should be briefly discussed for its role to pattern the cell adhesive substance (through ink transferring).

Line 14, Page 3 – why did the authors use “on the other hand”?

Line 16 – Line 18, Page 3 – “In the … into the mold” should be whole rewritten.

Line 4, Page 4 – “… the viscosity large …” should be rewritten.

Line 13, Page 18 – should be “slightly”

…

Last not the least, the Introduction part should be reorganized as well as the English for the whole manuscript should be proofread carefully.

2. What is “analogous gas – based cell repellent surface patterning method”? The manuscript should briefly mention about in the introduction.

3. Line 13, Page 4: “To cope with the viscosity issue …” is quite unsmooth from the aforementioned background. I did not see any viscosity impacts before.

4. The quality of figures should be improved with higher resolution.

5. I did not quite understand the description in Line 2, Page 6: “diffusion of air through the walls …” there is vacuum pressure in the microchannel. Why did the author say “diffusion of air … into the bulk PDMS”.

6. PLOS authors have the option to publish the peer review history of their article (what does this mean?). If published, this will include your full peer review and any attached files.

Reviewer #1: No

Reviewer #2: No

Reviewer #3: No

---

## [Author Response · Author response to Decision Letter 0]

14 Feb 2020

Dear Reviewers,

Thank you very much for reviewing the manuscript carefully and giving many important comments.

We attached our responses to your comments in the uploads files.

Best regards,

Yo Tanaka

---

## [Decision Letter · Decision Letter 1]

17 Apr 2020

Flow analysis on microcasting with degassed polydimethylsiloxane micro-channels for cell patterning with cross-linked albumin

PONE-D-19-34244R1

Dear Dr. Tanaka,

We are pleased to inform you that your manuscript has been judged scientifically suitable for publication and will be formally accepted for publication once it complies with all outstanding technical requirements.

With kind regards,

Talib Al-Ameri, Ph.D

Academic Editor

PLOS ONE

Additional Editor Comments (optional):

Reviewers' comments:

Reviewer's Responses to Questions

**Comments to the Author**

1. If the authors have adequately addressed your comments raised in a previous round of review and you feel that this manuscript is now acceptable for publication, you may indicate that here to bypass the “Comments to the Author” section, enter your conflict of interest statement in the “Confidential to Editor” section, and submit your "Accept" recommendation.

Reviewer #1: All comments have been addressed

Reviewer #2: All comments have been addressed

Reviewer #3: (No Response)

2. Is the manuscript technically sound, and do the data support the conclusions?

Reviewer #1: Yes

Reviewer #2: Yes

Reviewer #3: Yes

3. Has the statistical analysis been performed appropriately and rigorously? 

Reviewer #1: Yes

Reviewer #2: Yes

Reviewer #3: (No Response)

4. Have the authors made all data underlying the findings in their manuscript fully available?

Reviewer #1: Yes

Reviewer #2: Yes

Reviewer #3: Yes

5. Is the manuscript presented in an intelligible fashion and written in standard English?

Reviewer #1: Yes

Reviewer #2: Yes

Reviewer #3: No

6. Review Comments to the Author

Reviewer #1: (No Response)

Reviewer #2: The author has addressed all comments. The revised manuscript has met the standard of the journal. I recommended it for pubilication.

7. PLOS authors have the option to publish the peer review history of their article (what does this mean?). If published, this will include your full peer review and any attached files.

Reviewer #1: No

Reviewer #2: Yes: Liguo Qin

Reviewer #3: No

---

## [Editor Report · Acceptance letter]

28 Apr 2020

PONE-D-19-34244R1 

Flow analysis on microcasting with degassed polydimethylsiloxane micro-channels for cell patterning with cross-linked albumin 

Dear Dr. Tanaka:

I am pleased to inform you that your manuscript has been deemed suitable for publication in PLOS ONE. Congratulations! Your manuscript is now with our production department. 

With kind regards,

on behalf of

Dr. Talib Al-Ameri 

Academic Editor

PLOS ONE